# A Randomized Controlled Study to Evaluate the Safety and Reactogenicity of a Novel rVLP-Based Plant Virus Nanoparticle Adjuvant Combined with Seasonal Trivalent Influenza Vaccine Following Single Immunization in Healthy Adults 18–50 Years of Age

**DOI:** 10.3390/vaccines8030393

**Published:** 2020-07-20

**Authors:** Joanne Langley, Elodie Pastural, Scott Halperin, Shelly McNeil, May ElSherif, Donna MacKinnon-Cameron, Lingyun Ye, Cécile Grange, Valérie Thibodeau, Jean-François Cailhier, Rejean Lapointe, Janet McElhaney, Luis Martin, Marilène Bolduc, Marie-Eve Laliberté-Gagné, Denis Leclerc, Pierre Savard

**Affiliations:** 1Canadian Center for Vaccinology (Dalhousie University, IWK Health Centre, Nova Scotia Health Authority), 5850 University Avenue, Halifax, NS B3K 6R8, Canada; scott.halperin@dal.ca (S.H.); shelly.mcneil@nshealth.ca (S.M.); May.Elsherif@iwk.nshealth.ca (M.E.); Donna.MacKinnon-Cameron@dal.ca (D.M.-C.); Lingyun.Ye@iwk.nshealth.ca (L.Y.); 2Department of Pediatrics, Faculty of Medicine, Dalhousie University, 5850 University Avenue, Halifax, NS B3K 6R8, Canada; 3Pan-Provincial Vaccine Enterprise (PREVENT), 120 Veterinary Road, Saskatoon, SK S7N 5E3, Canada; elodie.pastural@usask.ca (E.P.); luis.martin@utoronto.ca (L.M.); 4Microbiology & Immunology, Faculty of Medicine, Dalhousie University, 5850 College Street, Halifax, NS B3H 4H7, Canada; 5Centre de Recherche du Centre Hospitalier de l’Universite de Montreal (C.R.H.U.M.), 900 rue St-Denis, Tour Viger, Montreal, QC H2X 0A9, Canada; grange.cme@gmail.com (C.G.); valeriesarati@hotmail.com (V.T.); jf.cailhier@umontreal.ca (J.-F.C.); rejean.lapointe@umontreal.ca (R.L.); 6Institut du cancer de Montréal, Montréal, 900 St-Denis Street, 10th floor, Montreal, QC H2X 0A9, Canada; 7Département de Médecine, Faculté de Médecine, Université de Montréal, 2900 Edouard Montpetit Blvd, Montreal, QC H3T 1J4, Canada; 8Northern Ontario School of Medicine, 41 Ramsey Lake Health Centre, Thunder Bay, ON P3E 5J1, Canada; jmcelhaney@hsnri.ca; 9Department of Microbiology, Infectiology and Immunology, Laval University, 1050, avenue de la Médecine, bureau 4633, QC G1V 0A6, Canada; marilene.bolduc@crchudequebec.ulaval.ca (M.B.); Marie-Eve.L-Gagne@crchudequebec.ulaval.ca (M.-E.L.-G.); Denis.Leclerc@crchudequebec.ulaval.ca (D.L.); 10Department of Molecular Biology, Medical Biology and Pathology, Laval University, 2705 boulevard Laurier P-09835, QC PQ G1V 4G2, Canada; Pierre.Savard@fmed.ulaval.ca

**Keywords:** influenza vaccines, immunogenicity, adjuvants

## Abstract

Inactivated influenza vaccines efficacy is variable and often poor. We conducted a phase 1 trial (NCT02188810), to assess the safety and immunogenicity of a novel nanoparticle Toll-like receptor 7/8 agonist adjuvant (Papaya Mosaic Virus) at different dose levels combined with trivalent influenza vaccine in healthy persons 18–50 years of age. Hemagglutination-inhibition assays, antibody to Influenza A virus nucleoprotein and peripheral blood mononuclear cells for measurement of interferon-gamma ELISPOT response to influenza antigens, Granzyme B and IFNγ:IL-10 ratio were measured. The most common adverse events were transient mild to severe injection site pain and no safety signals were observed. A dose-related adjuvant effect was observed. Geometric mean hemagglutination-inhibition titers increased at day 28 in most groups and waned over time, but fold-antibody responses were poor in all groups. Cell mediated immunity results were consistent with humoral responses. The Papaya Mosaic Virus adjuvant in doses of 30 to 240 µg combined with reduced influenza antigen content was safe with no signals up to 3 years after vaccination. A dose-related adjuvant effect was observed and immunogenicity results suggest that efficacy study should be conducted in influenza antigen-naïve participants.

## 1. Introduction

Influenza virus causes seasonal epidemics of respiratory illness with a heavy burden in the young, the older adult and those with a wide range of chronic conditions such as heart and lung disease, diabetes and immunocompromise [1]. Although use of influenza vaccine reduces the societal burden of health care visits, hospitalization and death and is accepted as the primary public health intervention worldwide to prevent influenza [2], the efficacy of seasonal influenza vaccines is suboptimal [3]. Use of adjuvants is one of several strategies being pursued to enhance immune responses and hence clinical protection against influenza [4,5]. Oil in water adjuvanted pH1N1 2009 influenza vaccines for example, with reduced influenza antigen as low as 3.75 micrograms, were highly effective [6]. 

An ideal adjuvant would promote immune response to a vaccine antigen, have minimal reactogenicity and meet public and regulatory standards for safety. Virus-like nanoparticles (VLPs) vaccines have been used to promote robust and durable immune responses to Human Papilloma Virus, Hepatitis B Virus [7] and pathological aggregated tau protein (Qb bacteriophage) [8]. VLPs contain no genetic material and thus are not infectious. The highly ordered structure of VLP is known to effectively present highly ordered foreign epitopes and induce a T-cell independent antibody response [9,10,11]. Inclusion in their structure of an endosomal toll like receptor (TLR) agonist allows targetting its delivery to immune cells only and was shown to enhance immune response to conjugated vaccine antigens [12]. In pre-clinical studies, a VLP-based adjuvant derived from the coat protein of the non-pathogenic Papaya Mosaic Virus (PapMV) was shown to enhance both humoral and cellular mediated immune response to TIV [13]. In addition, the fusion of peptide antigens directly to the surface of the PapMV VLP was shown to enhance significantly both humoral [14,15] and cellular mediated immune response to the co-presented antigens [16]. PapMV VLPs were shown to enhance microbial antigenic presentation on major histocompatibility complexes I (MHC class I), an efficient mechanism to mount a cytotoxic T lymphocyte (CTL) response [17]. Vaccination of mice with PapMV VLPs combined with viral proteins [13] or conjugated to viral epitopes at their surface have been shown to induce protection through antigen-specific antibodies [14,15,18].

In a phase 1, first-in-humans randomized controlled trial, we assessed the safety and immunogenicity of PapMV VLP based adjuvant at four different dose levels combined with commercially available inactivated trivalent influenza vaccine (TIV) in healthy adults. 

## 2. Materials and Methods 

This was a phase 1, first-in-humans, randomized (5:1), observer-blind, controlled study of the safety and immune response to intramuscular injection of one of four dose levels of PapMV recombinant VLP (Papaya Mosaic Virus (PapMV) Adjuvant Long lasting immune response, a term used for the PapM rVLP technology, PAL adjuvant or “PAL”) combined with one of two dose levels of a licensed trivalent inactivated influenza vaccine in healthy adults. The control vaccine was a standard dose of TIV. The PAL-adjuvanted vaccines contained various doses of PAL and reduced TIV content in order to assess if the adjuvant compensated for dose sparing. The full dose TIV was chosen based on regulatory advice. 

The study was initiated on 22 July 2013, day 360 visits concluded on 16 March 2015 and the last day 1095 visit occurred on 12 July 2017. The study (registered under ClinicalTrials.gov identifier NCT02188810) was undertaken in compliance with Good Clinical Practice guidelines, the Declaration of Helsinki and national regulatory requirements and was approved by local or regional institutional review boards at each study site. 

### 2.1. Participants

Participants were enrolled at one study center (the Canadian Center for Vaccinology in Halifax, Canada). Eligible participants were 18 to 50 years of age inclusive, in good general health as determined by history, physical examination and screening laboratory tests (hematological and biochemical), gave written informed consent and if female and of child-bearing potential and heterosexually active, were not pregnant and were practicing adequate contraception for 30 days prior to injection until 180 days after injection. Exclusionary criteria were confirmed or suspected immunosuppressive or immunodeficient condition or family history of same, evidence of Hepatitis B or C or Human Immunodeficiency Virus (HIV) infection, autoimmune disease, malignancy or lymphoproliferative disorder within previous 5 years, substance abuse, current pregnancy or lactation, allergy to any study product, use of any investigational or non-registered product (drug or vaccine) within 28 days preceding the dose of study product or of any blood product within 3 months prior to study vaccine or planned administration during the study or planned use during the study period, participation in another study or receipt of seasonal influenza vaccine within 120 days before injection to 121 days after.

### 2.2. Vaccine

Participants received one of 4 presentations of the PAL adjuvant (30 µg, 60 µg, 120 µg or 240 µg of the rVLP proteins) combined with a half dose of a seasonal TIV (~7.5 µg HA of each of three influenza strains) or 240 µg of the PAL adjuvant, combined with a quarter dose of the TIV (~3.5 µg HA of each of the three influenza strains) (Figure 1) or 0.5 mL TIV (control group). The PAL adjuvant in its final formulation is a liquid suspension of recombinant PapMV VLPs in 10 mM Tris solution (Millipore Sigma, Etobicoke, Canada) and is translucent to whitish in appearance. The PapMV rVLP is assembled onto a 1517 nucleotide-long RNA template. The final liquid suspension is known as FB-631. 

The TIV (FLUVIRAL^®^, GlaxoSmithKline Inc, Kirkland PQ, Canada) contained the World Health Organization recommended strains for the 2013–2014 season—A/California/7/2009 (H1N1) pdm09-like virus (A/California/7/2009 NYMC X-179A), A(H3N2) virus antigenically like the cell-propagated prototype virus A/Victoria/361/2011(A/Texas/50/2012 NYMC X-223A) and B/Massachusetts/2/2012-like virus (B/Massachusetts/2/2012 NYMC BX-51B). Each adult dose of 0.5 mL FLUVIRAL^®^ contains 15 μg HA of each influenza strain present in the vaccine. 

All products were stored under refrigeration (2–8 °C). An unblinded pharmacist prepared the study vaccine of the day of injection, affixing an obscuring label to the vaccine syringe and had no other role in the study. 

Each participant received a single dose of study vaccine as a 1mL intramuscular injection on day 0.

### 2.3. Study Procedures

At the screening visit, after the consent process, a medical history and physical examination were performed and safety blood samples were obtained. Eligible participants returned for Visit 1 within 30 days of the screening visit. 

Study group assignment was allocated randomly using a computer-generated step specific randomization list using PROC PLAN in SAS v 9.4, by the data manager. The overall randomization ratio was 5:1. There were four dose escalation steps (1–4) by which higher doses of the PAL adjuvant were given to subsequent participants. Within each step, the first three participants were randomly assigned 2:1 to the test article or control group and each injection was given four hours apart with a maximum of two participant injections per day. The remaining seven subjects in that step were randomized 6:1 to the test article or control groups and received vaccine at least 60 min apart. For the 4th and last step the first three participants were randomized 1:1:1 and the final 15 participants were randomized 7:7:1. Injection of participants in the subsequent step was contingent on approval of an external Safety Review Committee, which reviewed blinded Day 7 safety data of the prior step. 

Participants attended study sites at the screening visit and on Days 0, 7, 28, 120 and 180. Phone calls to participants by study staff to inquire about interim changes in health occurred on Days 360, 730 and 1095. 

At the vaccination visit participants were instructed on the use of a diary card for recording any solicited injection site or general adverse events (AEs), any unsolicited AEs and concomitant medications for seven days and to bring the diary to the next visit. A second diary card was provided on Day 7 to record unsolicited AEs and concomitant medications to Day 28. At each study visit participants were asked if there were any changes in their health or AEs since the last visit. 

### 2.4. Outcomes

The primary outcome was the assessment of safety and reactogenicity from vaccination up to Day 180. Solicited local AEs (pain, redness and swelling) and general AEs (drowsiness, fever, nausea, diarrhea, vomiting and generalized muscle aches) were collected by the participant on diary cards on the day of vaccination and for the following 6 days. Unsolicited AEs were collected on diary cards on the day of vaccination and for the next 27 days. Unsolicited AEs were classified according to the Medical Dictionary for Regulatory Activities (MeDRA^®^, International Conference on Harmonization).

Hematological (hemoglobin level, white blood cell count, lymphocyte, neutrophil, eosinophil and platelet count) and biochemical (alanine amino-transferase, aspartate amino-transferase and creatinine) safety assessments were performed on Day 0, 7 and 28. Grading of intensity of laboratory parameters was based on United States Food and Drug Administration guidance [19]. Hematological and biochemical tests and screening tests for HIV, Hepatitis B and C were performed at the local diagnostic laboratory. 

SAEs, AEs leading to study withdrawal and potentially immune mediated AEs were recorded throughout the study. 

The intensity grading scheme for solicited AEs is seen in Appendix A. Unsolicited AEs were assigned to the categories of mild (easily tolerated by the participant, causing minimal discomfort and not interfering with everyday activities, Grade 1), moderate (sufficiently discomforting to interfere with normal everyday activities, Grade 2) or severe (preventing normal, everyday activities, Grade 3), by the investigator. 

Study holding rules were in place and all SAEs were reported to the Safety Review Committee (SRC) in addition to local, institutional and national authorities, as required. The SRC reviewed safety data at each successive dose level and approved moving to the next dose level.

### 2.5. Immunogenicity 

Secondary objectives of the study were to evaluate humoral immune responses prior to receipt of study vaccine (Day 0) and after a single dose of study vaccine, on days 28, 120 and 180. Blood samples were collected on these days. A Day 120 time point was chosen to assess duration of immune responses to vaccines, in case participants chose to receive the standard seasonal influenza vaccine for the 2014–2015 season before the Day 180 endpoint. 

Laboratory tests were performed at the Canadian Center for Vaccinology laboratories, unless otherwise stated. Hemagglutination-inhibition assays (HI) to titer HA antibodies against the three influenza viruses contained in the TIV were performed using turkey red blood cells and standardized and validated procedures [20]; these were performed in duplicate on a separate day. Influenza viruses used were H1N1 (A/California/07/2009); H3N2 (A/Victoria/361/2011) and B (B/Massachusetts/02/2012), provided by Charles River Laboratories, International, Inc. A blinded sample of 20 serum specimens were re-tested for HI titers at the British Columbia Communicable Disease Centre laboratory (BC, Canada). 

Antibody to Influenza A virus nucleoprotein (NP) was performed using an enzyme-linked immunosorbent assay (ELISA) (Virusys Corp., Taneytown, MD, USA). 

Human peripheral blood mononuclear cells were challenged with the three live influenza virus strains described above, to obtain cell culture supernatants for cytokine assays (interferon-gamma:interleukin-10; IFN-γ:IL-10 ratio) and cell culture lysates for Granzyme B levels. Granzyme B levels were determined using the methods of McElhaney et al. [21]. IFN-γ and IL-10 were determined using a magnetic bead multiplex assay (Milliplex^®^, Millipore (Canada) Ltd., Etokicoke, Canada) on a Luminex MAR^®^ platform (ThermoFisher Scientific, Mississauga, Ontario, Canada). The IFNγ ELISPOT was conducted using no stimulation (control) or one of five peptide stimulants (PMA/Ionomycin, Pan-MHC Cytomegalovirus-Epstein Barr virus-influenza (CEF), PepMix Influenza A CD4 MP1/Ann Arbor (H2N2), PepMix Influenza A CD4 (NP/Ann Arbor (H2N2), ProMix Influenza peptide pool and TIV) at the Centre Hospitalier Universitaire de Montréal (CHUM, PQ, Canada) using the ImmunoSpot Series 3B Analyzer [22]. 

### 2.6. Statistical Analysis

This was a first-in-human study and therefore there were no previous estimates of the frequency of AEs or of immune responses. Given the sample size of 40 and 8 for the treatment and control groups respectively, AEs occurring at a rate of 5% or greater in the treatment group would be estimated to be detected with a probability exceeding 0.81, while AEs in the control groups occurring at a rate of 20% or greater would be detected with a probability exceeding 0.83.

Data analysis for the period to day 180 was initiated once all safety and HAI immune assessments for all participants were completed. The treatment status of participants was unblinded by group assignment in order for the data analysis team to complete these analyses. The treatment allocation of each individual participant remained blinded to the participant and the study personnel until all Day 1095 visits were complete and the database was locked. 

The safety analysis was performed on the safety population, which is all participants who received at least one study vaccine. Baseline comparability of treatment groups was assessed using binomial estimates and Fisher’s exact tests for binary variables; and *t*-tests and confidence intervals for continuous variables. For analysis of proportions, binomial point estimates and exact binomial confidence intervals were calculated for each group and differences between groups compared using Fisher’s exact tests. All statistical tests were 2-sided with a Type I error of 5%. 

The analysis of continuous variables consisted of point estimates and interval estimates for means and differences between groups were assessed using *t*-tests and analysis of variance. Missing values were not included in the analyses and there was no imputation of missing values. No adjustment was made for multiple comparisons.

Geometric mean antibody titers (GMTs) of HI titers for each influenza strain and mean GMT ratios (test strain/placebo) and their two side 95% confidence intervals were calculated by group. Results of cell-mediated immune tests on peripheral blood mononuclear cells were described [23].

## 3. Results

All but eight screened participants met study criteria and were enrolled, randomized and received study vaccine (Figure 1). There were no withdrawals. Demographic characteristics of participants according to study groups are seen in Table 1. 

### 3.1. Safety

The most common solicited local AEs in the 7 days after vaccine receipt was mild to severe pain at the injection site, which occurred in 62.5% to 87.5% of PAL-TIV vaccine recipients in each group (Figure 2). No participant was febrile in the seven-day period after vaccination. Muscle aches of grade 1 to 2 severity occurred in all groups. 

There was one SAE deemed unrelated to vaccine by the blinded investigator. There were 2 cases of potentially immune mediated events diagnosed after vaccine receipt (psoriasis) but both were determined by the blinded investigator to have had onset prior to study enrolment. No holding rules were met at any point during the study. Most hematologic or chemistry values remained within the normal local reference ranges throughout the study; any values outside these ranges were deemed not clinically significant by the blinded investigator (s).

### 3.2. Immune Responses 

All participants had HI and anti-NP titers at baseline. HI Geometric Mean Titer Ratios (GMTRs) increased at Day 28 post vaccine in most groups and then waned by the day 120 visit (Figure 3). (After the day 120 visit participants were free to be vaccinated with seasonal TIV as recommended by local public health; 67% (32/48) of participants did so before day 180.) Higher doses of PAL adjuvant (240 µg) were associated with higher HI GMTs. Participants in the study groups with the lowest dose of PAL (30 or 60 μg PAL) had the lowest HI GMTs to all three influenza strains. Responses to the H1N1 and H3N2 strains were highest in the 240 μg PAL containing vaccines (0.25 mL and 0.125 mL TIV).

HI fold-antibody (FAR) responses to TIV strains were poor in all groups (≤37.5% of participants/group had 4-FAR to any strain) (Appendix A). GMTs to NP at Day 28 post vaccine were increased above baseline in all groups and were highest in the control group (data not shown). 

Geometric mean granzyme B levels to all influenza strains were present prior to vaccine receipt in all study groups and were sustained throughout the observation period (Appendix A). The highest responses post vaccine were seen in the control group (standard dose of TIV). 

Geometric mean interferon-gamma to interleukin-10 ratios (IFNγ:IL-10) were highest in the control group across all three strains (Appendix A). Geometric mean IFNγ secreted in response to peptide stimulants was not significantly different across study groups. When results were analyzed according to whether or not the participant opted to receive a standard dose seasonal TIV after day 120, no differences in results were observed (data not shown).

## 4. Discussion

In this first-in-human controlled trial of a novel rVLP-based plant virus nanoparticle adjuvant co-presented with various concentrations of a seasonal influenza vaccine, we observed an adjuvant effect, where participants receiving larger doses (240 and 120 mcg) of PAL generally had higher GMTs that were more similar to the full dose of TIV without adjuvant. The adjuvant effect was generally dose-related, with the highest dose of PAL adjuvant being associated with higher GMTs. In preclinical studies in mice and ferrets, the PAL adjuvant triggered both humoral and broad cell-mediated immune responses to influenza antigens which were long lasting [12,13,18]. In pre-clinical challenge studies this response correlated with heterosubtypic protection from lung influenza infection [13,18]. Studies in animal models were necessarily done in influenza-naïve subjects and we speculate that this may explain the somewhat discrepant findings between the human and non-human studies. The human immune response to influenza is known to be shaped by imprinting, such that epitopes presented in early exposures determine the immune response in subsequent exposures [24,25] If this is correct, an adjuvant co-presented with influenza antigen may be unable to surmount priming effects; this may explain why low doses of TIV combined with the PAL adjuvant did not result in titers higher than standard dose TIV. The fact that participants who received a new full dose of TIV after Day 120 did not show higher levels of humoral or cellular response at Day 180 supports this interpretation. We suggest that future studies in humans use antigen-naïve participants and consider targets other than influenza, since influenza priming generally occurs in childhood. 

Since the PAL adjuvant is novel and its effect is as a TLR-7/8 ligand, the study was designed to carefully monitor participant safety and continue surveillance for unexpected adverse events for three years after vaccine receipt. Over this prolonged period we did not observe any untoward effects or safety signals. The most common solicited AEs were similar to other inactivated vaccines, including transient mild to moderate pain at the injection sites, drowsiness and generalized muscle aches. Although the sample size of this study is small (n = 48) the safety profile over a prolonged period is reassuring. 

In summary, we conducted a dose-ranging first-in-human study of a novel TLR7 adjuvant derived from a non-pathogenic plant virus (PapMV) co-presented with a seasonal influenza vaccine. All study vaccines were safe with acceptable reactogenicity and no unexpected AEs were seen three years after vaccine receipt. An adjuvant effect was observed and immunogenicity results suggest that, using the same biological markers, an efficacy study should be conducted in antigen-naïve participants to better assess the role of this adjuvant. 

## 5. Conclusions

Inactivated influenza vaccines efficacy is variable and often poor. In a first in humans trial we assessed the safety and immunogenicity of a novel nanoparticle Toll-like receptor 7/8 agonist adjuvant (Papaya Mosaic Virus) at different dose levels combined with trivalent influenza vaccine in healthy persons 18–50 years of age. The most common adverse events were transient mild to severe injection site pain and no safety signals were observed after three years of follow up. A dose-related adjuvant effect was observed. Geometric mean hemagglutination-inhibition titers increased at day 28 in most groups and waned over time, but fold-antibody responses were poor in all groups. Cell mediated immunity results were consistent with humoral responses. An adjuvant effect was observed and immunogenicity results suggest that, using the same biological markers, an efficacy study should be conducted in antigen-naïve participants to better assess the role of this adjuvant.

## Figures and Tables

**Figure 1 vaccines-08-00393-f001:**
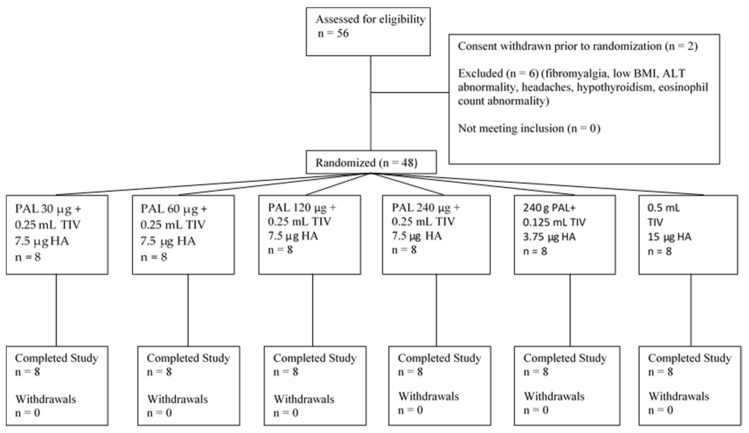
Participant Flow.

**Figure 2 vaccines-08-00393-f002:**
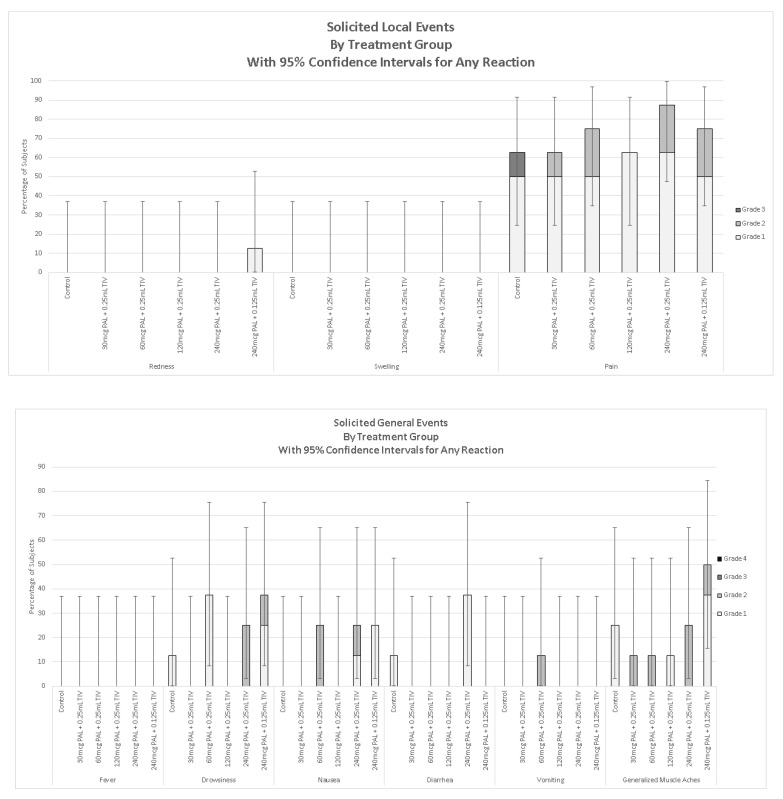
Solicited local and systemic adverse events in the seven days after vaccination. Legend. Control–Trivalent inactivated influenza vaccines (TIV) 0.5 mL; 30 µg PAL adjuvant and 0.25 mL TIV; 60 µg PAL adjuvant and 0.25 mL TIV; 120 µg PAL adjuvant and 0.25 mL TIV; 240 µg PAL adjuvant and 0.25 mL TIV, 240 µg of PAL adjuvant and 0.125 mL TIV. Redness or swelling grade 1—>20–≤50 mm diameter; Grade 2—>50–≤100 mm diameter, Grade 3—>100 mm; Pain at injection site Grade 1—mild, neither interfering nor preventing normal every day activities, Grade 2—moderate, painful when limb is moved and interferes with normal every day activities, Grade 3—severe, significant pain at rest. Prevents normal every day activities. Error bars show 95% confidence intervals.

**Figure 3 vaccines-08-00393-f003:**
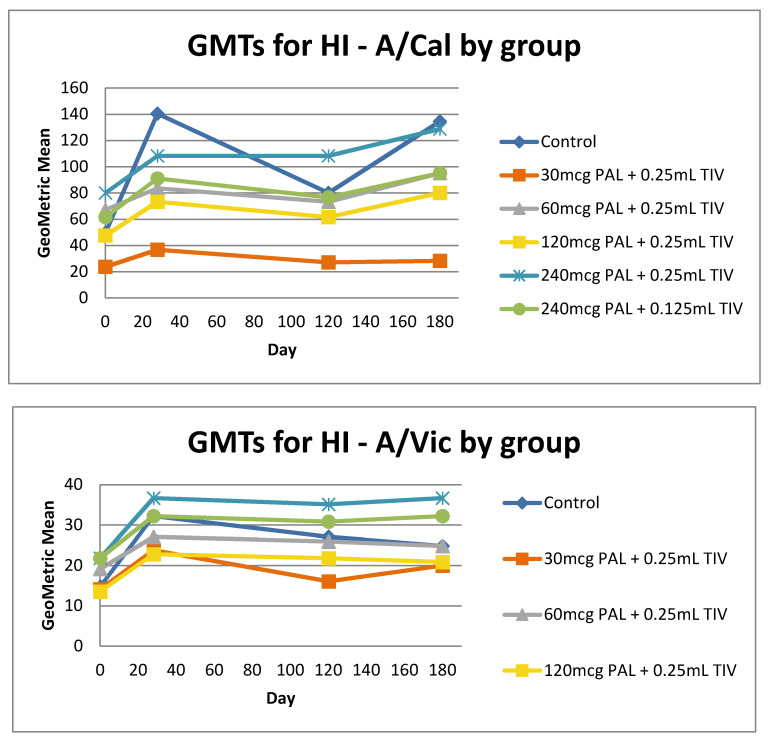
Haemagglutinin Inhibition (HI) Geometric Mean Titers (GMT) to PAL-adjuvanted Trivalent seasonal influenza vaccine (TIV) from Day 0 to 180 in healthy young adults. Legend. A/California–Influenza A/California/7/2009 (H1N1) pdm-like virus, A/Victoria-Influenza A/Victoria/361/2011, B/Mass-B/Massachusetts/2/2012-like virus. Note, after day 120 participants were free to be vaccinated with the seasonal influenza vaccine as recommended by local public health authorities.

**Table 1 vaccines-08-00393-t001:** Participant Demographics.

Characteristic	PAL 30 μg + 0.25 mL TIV	PAL 60 μg + 0.25 mL TIV	PAL 120 μg + 0.25 mL TIV	PAL 240 μg + 0.25 mL TIV	PAL 240 μg + 0.125 mL TIV	0.5 mL TIV (Control)
n	8	8	8	8	8	8
Mean age in years (SD)	44.1 (3.6)	35.6 (7.7)	33.4 (6.5)	39.0 (7.4)	35.5 (9.3)	42.5 (7.4)
Age Range	38–47	21–45	23–42	22–43	20–48	31–50
Gender–Female % (n)	62.5 (5)	75 (6)	62.5 (5)	87.5 (7)	37.5 (3)	75 (6)
Ethnicity %						
Caucasian	100	100	100	100	87.5 (7)	87.5 (7)
Asian	-	-	-	-		1 (12.5)
Other	-	-	-	-	1 (12.5)	-

n = number.

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
