# Peer review of "A Randomized Controlled Study to Evaluate the Safety and Reactogenicity of a Novel rVLP-Based Plant Virus Nanoparticle Adjuvant Combined with Seasonal Trivalent Influenza Vaccine Following Single Immunization in Healthy Adults 18–50 Years of Age"

_vaccines, 2020, doi:10.3390/vaccines8030393_

Round 1
Reviewer 1 Report
Dear Authors,
I appreciate your work on"evaluate the safety and reactogenicity of a novel rVLP-based plant virus nanoparticle adjuvant combined with seasonal 4 trivalent influenza vaccine". However, i have few concern which need to be addressed;
- In Figure 3, why the control showed higher levels in GMTs for HI -A/Cal by group and HI -B/Mass by group as compared to HI -A/Vic by group?
- whats the toxicity levels of Virus-like nanoparticles to B and T cells?
- how specific is a novel rVLP-based plant virus nanoparticle adjuvant combined with seasonal trivalent influenza vaccine following single immunization?
Thanks,
LR
Author Response
I appreciate your work on "evaluate the safety and reactogenicity of a novel rVLP-based plant virus nanoparticle adjuvant combined with seasonal 4 trivalent influenza vaccine". However, i have few concern which need to be addressed;
- In Figure 3, why the control showed higher levels in GMTs for HI -A/Cal by group and HI -B/Mass by group as compared to HI -A/Vic by group?
Response
Figure 3 shows the Haemagglutinin Inhibition Geometric Mean Titers (GMT) to PAL-adjuvanted Trivalent seasonal influenza vaccine from Day 0 to 180 in healthy young adults. As noted in the results, the GMTS increased at day 28 post vaccine and then waned by the day 120 visit. The reviewer suggests that the GMTs are higher in the control group against A/Cal and B/Mass than they are for A/Vic. We did not make an interpretation as to the relative increases across study groups against the three antigens as no hypothesis testing was planned or done in this phase 1 study in which the primary outcome was safety. The sample sizes in each group are small (n=8).
We have not amended the text of the manuscript as the methods indicate that no formal comparisons were planned for this outcome.
- whats the toxicity levels of Virus-like nanoparticles to B and T cells?
Response
We have no evidence of direct toxicity on in vitro/in vivo T and B cells. However, preclinical in vivo data provide indirect evidence that there is no toxicity on the immune system of repeated FB-631 administration to rat and rabbits with a non-observable adverse effect level (NOAEL) of 0.9 and 1 mg/dose, respectively (study 1009-2591 and study 1009-2574). These studies included clinical pathology (hematology and clinical chemistry) and immunology assessments (total IgG and IgM). We also demonstrated that following administration of fluorescent-labelled FB-631 in the footpad of mice the proximal popliteal lymph node became fluorescent 24h later; then, the signal declined progressively over the subsequent 48h suggesting that the nanoparticles are rapidly degraded (Savard et al 2011; see ref. list). Finally, we also know that FB-631 do not enter into human B or T cells but is preferentially internalized by monocytes, macrophages and dendritic cells (antigen presenting cells). This observation makes it less likely that FB-631 can be toxic to B or T cells (Carignan et al., 2018; see ref list).
We have not amended the text of the manuscript as we would need to cite our own papers to do so. Some of the papers are referenced and the reader can find more data about preclinical (animal models) work there.
- how specific is a novel rVLP-based plant virus nanoparticle adjuvant combined with seasonal trivalent influenza vaccine following single immunization?
Response:
We take the reviewer’s question to inquire as to whether vaccinated participants would have a different or similar immune response to a heterologous influenza vaccine. This is an important question. We did not test heterologous antibody responses in this study so are not able to answer this question with the data from this study.
Reviewer 2 Report
This study presented by authors evaluated the efficacy and toxicity of Influenza antigen with different doses of antigen. The study has some merit but the design is very complex with too many groups which makes it hard to follow.My primary concern is the way data is presented and study design.
Figure 1 can be simplified. If all the participants completed study and none withdrew, it would be easier to remove the bottom half the figure to make it less complex. Why was control dose of HA chosen to be higher than all the other HA doses?
Those participants should have been chosen for this study that hadn't received a Flu vaccine for 1-2 years. Then the response is true and is not inhibited by the host.
Figure 2 is easy to understand, but if this is a distribution graph, I don't understand what the error bars indicate. They should put titles on the Y-axis, and panels should be labelled.
Figure 3, some of the group indicators are missing in each panel. I don't understand the titers, it seems the titers are very low and doesn't make sense. They should indicate how they calculate the end point titers. It seems control is doing better than the other doses, it makes no sense to use this adjuvant then.
Suppl. Fig 1: I don't understand this figure at all.
Suppl. Fig 2: Why are titers so low? Plot it as end point titers to make a sense of this figure. Again none of the doses with the adjuvant did anything at all.
Suppl. Fig 3-4: Nothing useful there as well.
Author Response
This study presented by authors evaluated the efficacy and toxicity of Influenza antigen with different doses of antigen. The study has some merit but the design is very complex with too many groups which makes it hard to follow. My primary concern is the way data is presented and study design.
Figure 1 can be simplified. If all the participants completed study and none withdrew, it would be easier to remove the bottom half the figure to make it less complex.
Response:
The figure follows the CONSORT template for reporting randomized controlled clinical trials, which is an internationally accepted standard < http://www.consort-statement.org/consort-statement/flow-diagram>. We prefer to use this international standard which strengthens consistency of clinical trial reporting, but if the editors prefer that we change the figure we can do so.
Why was control dose of HA chosen to be higher than all the other HA doses?
Response:
The control dose is the standard dose given for annual immunization. The aim of the study was to determine if the adjuvanting effect of PAL could compensate for the lower dose of HA. The study doses combined lower dose of influenza antigen content with varying doses of adjuvant to determine if the adjuvant could boost the immune response to make up for the lower content of influenza antigen. The full dose TIV control was recommended by the regulator. We have added the following text to the methods, paragraph 1:
“The control vaccine was a standard dose of TIV. The PAL-adjuvanted vaccines contained various doses of PAL and reduced TIV content in order to assess if the adjuvant compensated for dose sparing. The full dose TIV was chosen based on regulatory guidance.”
Those participants should have been chosen for this study that hadn't received a Flu vaccine for 1-2 years. Then the response is true and is not inhibited by the host.
Response:
We agree with the reviewer that it would have been ideal to have enrolled persons who had not had influenza vaccine at all or at least for many years prior. However, annual seasonal influenza vaccine is provided in universal programs in most parts of Canada, so it is broadly seen as the standard of care.
Figure 2 is easy to understand, but if this is a distribution graph, I don't understand what the error bars indicate. They should put titles on the Y-axis, and panels should be labelled.
Response:
Figure 2 shows the solicited local and systemic adverse events in the seven days after vaccination. The error bars are 95% confidence intervals. We have added this to the legend for the figure:
“Error bars show 95% confidence intervals.”
We have uploaded a revised figure with relabelled y and x axes.
Figure 3, some of the group indicators are missing in each panel. I don't understand the titers, it seems the titers are very low and doesn't make sense. They should indicate how they calculate the end point titers. It seems control is doing better than the other doses, it makes no sense to use this adjuvant then.
Response:
Figure 3 shows Figure 3 shows the Haemagglutinin Inhibition Geometric Mean Titers (GMT) to PAL-adjuvanted Trivalent seasonal influenza vaccine from Day 0 to 180 in healthy young adults. All six groups are indicated in the legend on the right of the panel. The reviewer may not be seeing the grey 60 mcg group line for B/Mass as it is very similar to the 30 and 120 mcg groups.
With regard to the low titers, the reviewer interprets the finding correctly. The methods for the HI titers are internationally accepted, were validated at another laboratory, and are referenced. We have added the following phrase (italics) to the discussion to emphasize this point:
The human immune response to influenza is known to be shaped by imprinting, such that epitopes presented in early exposures determine the immune response in subsequent exposures. If this is correct, an adjuvant co-presented with influenza antigen may be unable to surmount priming effects; this may explain why low doses of TIV combined with the PAL adjuvant did not result in titers higher than those elicited by standard dose TIV.
Suppl. Fig 1: I don't understand this figure at all.
Response:
Supplemental Figure 1 shows the percentage of participants with four-fold antibody rise (FAR) of HI titer between day 0 and 28, by study group. A 4-fold increase of HI is a correlate of protection for influenza virus. In the discussion we note that “HI fold-antibody (FAR) responses to TIV strains were poor in all groups (≤37.5% of participants/group had 4-FAR to any strain)”, and refer to this figure.
Suppl. Fig 2: Why are titers so low? Plot it as end point titers to make a sense of this figure. Again none of the doses with the adjuvant did anything at all.
Response:
Supplemental Figure 2 shows influenza A Anti-Nucleoprotein Antibody Geometric Mean Titers to Day 0 to 180. We have deleted the figure and indicated the findings with summary text in the results, saying “data not shown”.
Suppl. Fig 3-4: Nothing useful there as well.
Response:
Supplemental Figures 3 and 4 show the cell mediated immune responses (Granzyme B and IFN gamma:IL-10 ratios) to each vaccine. We think this data is consistent with the humoral responses and will be of interest to the reader, and so have not deleted it.
Reviewer 3 Report
This is a brief, but very well presented study examining safety of a nanoparticle adjuvanted influenza vaccine to determine safety and possible immune enhancement.
The study is logical and clearly presented. The conclusions are supported by the data.
There are only a few modifications needed to make the presentation even clearer:
- Abbreviations should not be presented in the Abstract but should be defined at first use in the main body. TIV is not defined until in one of the later figures.
- The legend to the GMT-A/Vic is missing some information.
- The first few lines of the Discussion could be better described. While the study did examine "various levels" of TIV, most of the study was designed to test various PAL levels. Also, please discuss implications of the half-TIV vs quarter-TIV, not just vs full-TIV.
- Supplementary Fig 1; the X-axis labels are too close together and should be better separated.
Author Response
Reviewer 3
This is a brief, but very well presented study examining safety of a nanoparticle adjuvanted influenza vaccine to determine safety and possible immune enhancement.
The study is logical and clearly presented. The conclusions are supported by the data.
There are only a few modifications needed to make the presentation even clearer:
- Abbreviations should not be presented in the Abstract but should be defined at first use in the main body. TIV is not defined until in one of the later figures.
Response:
Abbreviations have been deleted from the abstract, and introduced where they first appear in the paper.
- The legend to the GMT-A/Vic is missing some information.
Response:
The HI GMTs to each of the three strains are shown in Figure 3. Here is the current version of the legend which serves to explain the figure for each strain: “Legend. A/California – Influenza A/California/7/2009 (H1N1) pdm-like virus, A/Victoria – Influenza A/Victoria/361/2011, B/Mass – B/Massachusetts/2/2012-like virus. Note, after day 120 participants were free to be vaccinated with the seasonal influenza vaccine as recommended by local public health authorities.” We are unsure what information is missing but would be happy to revise the legend with further direction.
- The first few lines of the Discussion could be better described. While the study did examine "various levels" of TIV, most of the study was designed to test various PAL levels. Also, please discuss implications of the half-TIV vs quarter-TIV, not just vs full-TIV.
Response:
The text has been revised as follows:
“In this first-in-human controlled trial of a novel rVLP-based plant virus nanoparticle adjuvant co-presented with various concentrations of a seasonal influenza vaccine, we observed an adjuvant effect, where participants receiving larger doses (240 and 120 mcg) of PAL generally had higher GMTs that were more similar to the full dose of TIV without adjuvant.”
As well the following text has been added in italics,(in response to another reviewer):
If this is correct, an adjuvant co-presented with influenza antigen may be unable to surmount priming effects; this may explain why low doses of TIV combined with the PAL adjuvant did not result in titers higher than standard dose TI.V
- Supplementary Fig 1; the X-axis labels are too close together and should be better separated
Response:
A revised figure is attached in which the labels are more spread out.